# TIME BLINDNESS: WHY VIDEO-LANGUAGE MODELS CAN'T SEE WHAT HUMANS CAN?

## ABSTRACT

Recent advances in vision–language models (VLMs) have made impressive strides in understanding spatio-temporal relationships in videos. However, when spatial information is obscured, these models struggle to capture purely temporal patterns. We introduce **SpookyBench**, a benchmark where information is encoded solely in temporal sequences of noise-like frames, mirroring natural phenomena from biological signaling to covert communication. Interestingly, while humans can recognize shapes, text, and patterns in these sequences with over 98% accuracy, state-of-the-art VLMs achieve 0% accuracy. This performance gap highlights a critical limitation: an over-reliance on frame-level spatial features and an inability to extract meaning from temporal cues. Overcoming this limitation will require novel architectures or training paradigms that decouple spatial dependencies from temporal processing. Our systematic analysis shows that this issue persists across model scales and architectures. We release SpookyBench to catalyze research in temporal pattern recognition and bridge the gap between human and machine video understanding. Dataset is available at this anonymous link: https://tinyurl.com/spooky-bench

## 1 INTRODUCTION

Large multimodal models have revolutionized visual understanding in both images (Liu et al., 2023; Wang et al., 2024b; Bai et al., 2025; Chen et al., 2024f; Deitke et al., 2024; Dai et al., 2024) and videos (Zhang et al., 2024b; Maaz et al., 2023; Ataallah et al., 2024; Weng et al., 2024; Wang et al., 2025). Recent Video-Vision Language Models (Video-VLMs) demonstrate impressive capabilities in various tasks, from action recognition (Wu et al., 2023; Kahatapitiya et al., 2024; Zhao et al., 2023) and visual question answering (Yu et al., 2023; Min et al., 2024; Zhong et al., 2022; Ayyubi et al., 2025; Park et al., 2024) to dense captioning (Qasim et al., 2025; Yang et al., 2023; Xu et al., 2024a; Kim et al., 2024; Chen et al., 2024d; 2025b; 2024a) and temporal grounding (Chen et al., 2024c; Wang et al., 2024a; Xu et al., 2024b). Despite this rapid progress, a fundamental limitation persists. These models excel at extracting spatial features from individual frames, but struggle with purely temporal reasoning (Cores et al., 2024; Cai et al., 2024; Li et al., 2024d), a capability that comes naturally to humans. This paper introduces **SpookyBench**, a novel benchmark designed to isolate and evaluate purely temporal understanding in video models by presenting information exclusively through temporal sequences where individual frames appear as noise. Although existing benchmarks test temporal reasoning alongside spatial understanding (Cai et al., 2024; Li et al., 2024e; Yang et al., 2025b; Li et al., 2024b), **SpookyBench** differs by completely eliminating spatial cues, forcing models to derive meaning solely from changes across frames. Current approaches to video understanding (Tang et al., 2023; Nguyen et al., 2024) typically follow a hierarchical paradigm: extract frame-level features using ViTs (Bertasius et al., 2021; Radford et al., 2021; Dosovitskiy et al., 2020), integrate these features temporally, and fuse them with language for downstream tasks (Zhang et al., 2024a; Li et al., 2024c; Wu et al., 2024; Wang et al., 2024c). This paradigm has yielded significant advances in general video understanding (Li et al., 2024a; Dubey et al., 2024; Tang et al., 2023; Nguyen et al., 2024). However, our findings reveal a critical blind spot: when information exists purely in the temporal domain without reliable frame-level features, state-of-the-art models fail catastrophically (Figure 1).

The inability to decode temporal patterns has significant implications for real-world applications. In nature, organisms such as fireflies communicate through precise temporal sequences of biolumi-

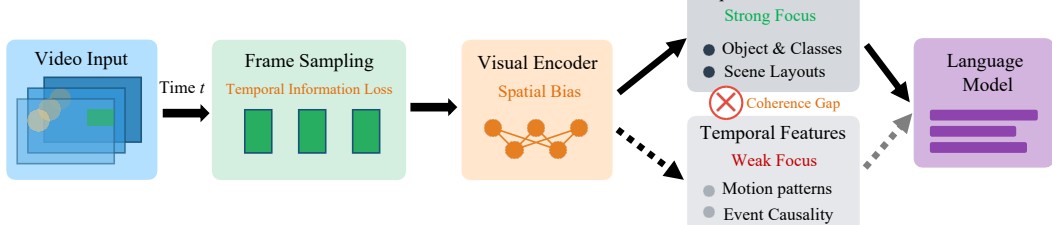

Figure 1: Illustration of the current video-language models' limitations: over reliance on spatial visual features within individual frame. Frame sampling results in temporal information loss, while the visual encoder exhibits a strong spatial bias. This creates a coherence gap (×) between well-represented spatial features (objects, scene layouts) and under-represented temporal features (motion patterns, causality), limiting video understanding capabilities.

nescence (Carlson & Copeland, 1985; Owens et al., 2022; Ramírez-Ávila et al., 2018), encoding information exclusively through timing rather than spatial arrangements. These natural examples demonstrate how temporal patterns can carry rich information even when individual observations contain minimal static content. Similarly, various human technologies from Morse code to digital communication protocols rely on temporal encoding, yet current Video-VLMs lack the fundamental mechanisms to process such information. The human visual system has evolved mechanisms for processing temporal information without relying solely on spatial cues. Neuroscience research has revealed that temporal processing is distributed across neural structures rather than centralized in a single area (Mauk & Buonomano, 2004), and the brain uses intrinsic network dynamics to perform temporal computations (Paton & Buonomano, 2018). Areas such as the parietal cortex integrate temporal information along with spatial and numeric magnitudes (Bueti & Walsh, 2009). Our experiments confirm humans' remarkable temporal perception: participants achieve over 98% accuracy on **SpookyBench** tasks without training. In stark contrast, our evaluation of 15 state-of-the-art Video-VLMs, including closed-source commercial systems such as GPT-4o (Hurst et al., 2024), and Gemini 2.0 Flash (DeepMind, 2025), reveals near-zero accuracy on these same tasks.

This striking performance gap persists across model architectures, parameter scales, and pre-training strategies. Models ranging from relatively compact systems (VideoLLaMA3-2B (Zhang et al., 2025)) to massive ones (GPT-4o (Hurst et al., 2024), Qwen-VL (Wang et al., 2024b)) all struggle with purely temporal patterns. Even models specifically designed for video understanding such as LongVLM (Weng et al., 2024), LLaVA-NeXT-Interleave (Li et al., 2024c), and InternVideo2.5 (Wang et al., 2025) exhibit minimal temporal pattern recognition capability.

Recent efforts to enhance temporal reasoning in Video-VLMs have explored various approaches. Models like TimeChat (Ren et al., 2024), Momentor (Qian et al., 2024), and VideoLLM (Wang et al., 2024e) incorporate specialized temporal modeling mechanisms, while ST-LLM (Liu et al., 2024b), TimeMaker (Chen et al., 2024c), and Grounded-VideoLLM (Wang et al., 2024a) focus on enhancing fine-grained temporal localization capabilities. However, our evaluation reveals that none of these approaches adequately addresses the fundamental challenge of extracting meaning from purely temporal patterns without reliable spatial features.

Our findings suggest that achieving human-like video understanding requires fundamentally rethinking how neural architectures process temporal information. Rather than treating temporal integration as secondary to spatial feature extraction, future models may need dedicated mechanisms for temporal pattern recognition, possibly drawing inspiration from cognitive neuroscience research on distributed neural timing mechanisms (Paton & Buonomano, 2018; Mauk & Buonomano, 2004) and specialized brain regions for temporal processing (Bueti & Walsh, 2009; Merchant et al., 2013). The substantial gap between human and machine performance on **SpookyBench** indicates that current architectures remain fundamentally "time-blind" despite their impressive performance on standard benchmarks. By exposing this critical limitation, we hope to inspire a new wave of research into temporal reasoning in Video-VLMs, bridging the gap between human and machine perception and enabling applications that rely on precise temporal understanding, from medical diagnostics to autonomous systems that must interpret subtle temporal cues in complex environments.

## 2 RELATED WORK

### 2.1 TEMPORAL REASONING IN VIDEO-VLMS

Transformer-based Video-Vision Language Models (Video-VLMs) have advanced through several architectural families, including LLaVA variants (Liu et al., 2023; Zhang et al., 2024b; Li et al., 2024c;a; Maaz et al., 2023; Liu et al., 2024a), the Qwen series (Wang et al., 2024b; Bai et al., 2025), and InternVL models (Chen et al., 2024g;f; Wang et al., 2025). Alternative approaches have explored dual encoders (Maaz et al., 2024), interleaved tokens (Ataallah et al., 2024; Zhu et al., 2023), compression techniques (Shen et al., 2024), and multimodal fusion (Zhang et al., 2024a; Wu et al., 2024; Wang et al., 2024c). Despite architectural diversity, these models consistently exhibit limited temporal reasoning, manifesting as hallucinations (Li et al., 2024b), grounding difficulties (Wang et al., 2024a), and a reliance on linguistic shortcuts (Ko et al., 2023) across action recognition (Wu et al., 2023; Kahatapitiya et al., 2024; Zhao et al., 2023), question answering (Yu et al., 2023; Min et al., 2024; Ayyubi et al., 2025), and captioning tasks (Yang et al., 2023; Kim et al., 2024; Chen et al., 2024a). Efforts to address these shortcomings; such as timestamp-aware encoding (Ren et al., 2024), segment-level reasoning (Qian et al., 2024), direct token processing (Liu et al., 2024b), temporal separation tokens (Chen et al., 2024c), specialized temporal streams (Wang et al., 2024a;e), and novel training paradigms (Zhang et al., 2024b; Yu et al., 2023; Tang et al., 2023; Nguyen et al., 2024); have shown incremental promise. However, these methods, and even specialized video architectures like VideoGPT+ (Maaz et al., 2024), TimeChat (Ren et al., 2024), LinVT (Gao et al., 2024), LongVLM (Weng et al., 2024), and Baichuan-Omni (Li et al., 2024f), still operate on a spatial-first paradigm where temporal understanding is secondary to spatial feature extraction.

The fundamental limitations of this spatial-first approach are increasingly evidenced by temporal understanding benchmarks. TemporalBench (Cai et al., 2024) reveals a significant gap between model and human performance, while TVBench (Cores et al., 2024), VITATECS (Li et al., 2024e), and Fateh et al. (Fateh et al., 2024) confirm that many datasets inadvertently reward spatial analysis over genuine temporal reasoning. Focused evaluations further target specific failures such as temporal hallucinations with VidHalluc (Li et al., 2024b), streaming video reasoning with SVBench (Yang et al., 2025b), and challenges in temporal location, object tracking, and anomaly detection with VideoVista (Li et al., 2024g). A critical and consistent finding across these analyses is that models—including video-specific ones like LLaVA-Video (Zhang et al., 2024b), Video-ChatGPT (Maaz et al., 2023), TemporalVLM (Fateh et al., 2024), and VidChain (Lee et al., 2025)—exploit spatial shortcuts to circumvent temporal reasoning (Wang et al., 2024a; Chen et al., 2024c; Li et al., 2024b; Ko et al., 2023). Our SpookyBench benchmark is designed to directly address this issue. By deliberately obscuring spatial information, it isolates temporal pattern recognition, forcing models to derive meaning solely from temporal dynamics. This approach provides a rigorous evaluation of the "time-blindness" in current architectures, exposing fundamental limitations that remain hidden in conventional assessments.

| Model | Direct Prompt | CoT | Params |
|---|---|---|---|
| Human Performance | 98.0% ± 0.6 | N/A | N/A |
| **Open-Source Models** | | | |
| VideoLLaMA3-7B (Zhang et al., 2025) | 0% ± 0.0 | 0% ± 0.0 | 7B |
| VideoLLaMA3-2B (Zhang et al., 2025) | 0% ± 0.0 | 0% ± 0.0 | 2B |
| TimeChat-7B (Ren et al., 2024) | 0% ± 0.0 | 0% ± 0.0 | 7B |
| MiniGPT4-Video (Ataallah et al., 2024) | 0% ± 0.0 | 0% ± 0.0 | 7B |
| MovieChat (Song et al., 2024) | 0% ± 0.0 | 0% ± 0.0 | 7B |
| Video-ChatGPT-7B (Maaz et al., 2023) | 0% ± 0.0 | 0% ± 0.0 | 7B |
| VideoGPT-plus-Phi3-mini-4k (Maaz et al., 2024) | 0% ± 0.0 | 0% ± 0.0 | 7B |
| VILA1.5-13BLin et al. (2024) | 0% ± 0.0 | 0% ± 0.0 | 13B |
| ShareGPT4Video-8B (Chen et al., 2024a) | 0% ± 0.0 | 0% ± 0.0 | 8B |
| VideoLLaMA2-7B (Cheng et al., 2024) | 0% ± 0.0 | 0% ± 0.0 | 7B |
| Video-LLaVA (Zhang et al., 2024b) | 0% ± 0.0 | 0% ± 0.0 | 7B |
| LLaVA-NeXT-Video (Li et al., 2024c) | 0% ± 0.0 | 0% ± 0.0 | 8B |
| InternVL2-40B (Chen et al., 2024f) | 0% ± 0.0 | 0% ± 0.0 | 40B |
| InternVL2-8B (Chen et al., 2024f) | 0% ± 0.0 | 0% ± 0.0 | 8B |
| InternVL2.5-78B (Chen et al., 2024e) | 0% ± 0.0 | 0% ± 0.0 | 78B |
| InternVL2.5-8B (Chen et al., 2024e) | 0% ± 0.0 | 0% ± 0.0 | 8B |
| InternVideo2.5-Chat-8B (Wang et al., 2025) | 0% ± 0.0 | 0% ± 0.0 | 8B |
| InternVideo2-Chat-8B (Wang et al., 2024d) | 0% ± 0.0 | 0% ± 0.0 | 8B |
| Qwen2-VL-2B-Instruct (Wang et al., 2024b) | 0% ± 0.0 | 0% ± 0.0 | 2B |
| Qwen2-VL-7B-Instruct (Wang et al., 2024b) | 0% ± 0.0 | 0% ± 0.0 | 7B |
| Qwen2-VL-72B-Instruct (Wang et al., 2024b) | 0% ± 0.0 | 0% ± 0.0 | 72B |
| Qwen2.5-VL-3B-Instruct (Bai et al., 2025) | 0% ± 0.0 | 0% ± 0.0 | 3B |
| Qwen2.5-VL-7B-Instruct (Bai et al., 2025) | 0% ± 0.0 | 0% ± 0.0 | 7B |
| Qwen2.5-VL-72B-Instruct (Bai et al., 2025) | 0% ± 0.0 | 0% ± 0.0 | 72B |
| **Closed-Source Models** | | | |
| Gemini 1.5 Pro (Team et al., 2024) | 0% ± 0.0 | 0% ± 0.0 | N/A |
| Gemini 2.0 FlashDeepMind (2025) | 0% ± 0.0 | 0% ± 0.0 | N/A |
| GPT-4o (Hurst et al., 2024) | 0% ± 0.0 | 0% ± 0.0 | N/A |

Table 1: Benchmark results comparing model performance on `SpookyBench` across different prompting strategies along with model size. Human accuracy (98.0%) is the weighted average of accuracy across 3 different categories.

## 2.2 NEUROSCIENCE INSIGHTS ON TEMPORAL PROCESSING

Neuroscience research offers critical insights for addressing temporal limitations in Video-VLMs. Mauk and Buonomano (Mauk & Buonomano, 2004) established that temporal processing is distributed across neural structures through intrinsic circuit properties, contrasting with current Video-VLMs' sequential spatial processing. Biological systems span multiple granularities: cerebellum handles millisecond-to-second timing (Merchant et al., 2013); parietal cortex integrates temporal, spatial and numerical magnitudes (Bueti & Walsh, 2009); and neural patterns dynamically encode time through "population clocks" (Paton & Buonomano, 2018). Models could benefit from distributed temporal representations that evolve over time (Wittmann, 2009; Paton & Buonomano, 2018) rather than treating temporal integration as secondary. The performance gap on temporal tasks (Cai et al., 2024; Cores et al., 2024; Li et al., 2024b) and our SpookyBench findings demonstrate that current architectures lack mechanisms for processing purely temporal patterns—a natural capability in humans through neural systems representing time as intrinsic dynamics.

## 3 SPOOKYBENCH

We introduce **SpookyBench**, a novel synthetic dataset specifically designed to isolate and evaluate pure temporal understanding in video language models. The key innovation of our benchmark lies in its unique design: All meaningful information is encoded exclusively in the temporal domain through dynamic patterns of texts, images and video depth maps, while individual frames contain only structured noise. Our dataset is fundamentally different from the existing datasets used for training, fine-tuning, and evaluation of video-VLMs. Many state-of-the-art video language models employ advanced techniques, such as dynamic resolution strategies (Bai et al., 2025; Wang et al., 2024b; Chen et al., 2024f), specialized temporal encoding methods (Ren et al., 2024; Wang et al., 2024b; Bai et al., 2025), hierarchical token merging (Weng et al., 2024; Wang et al., 2025), and joint video-motion training frameworks (Chen et al., 2024b) to capture temporal dynamics. However, these methods still rely on spatial representations extracted from individual frames, which currently remain the only viable mechanism for inferring temporal information. In contrast, **SpookyBench** forces models to depend only on temporal cues, thereby creating the first benchmark that exclusively evaluates a model's ability to process and understand pure temporal information.

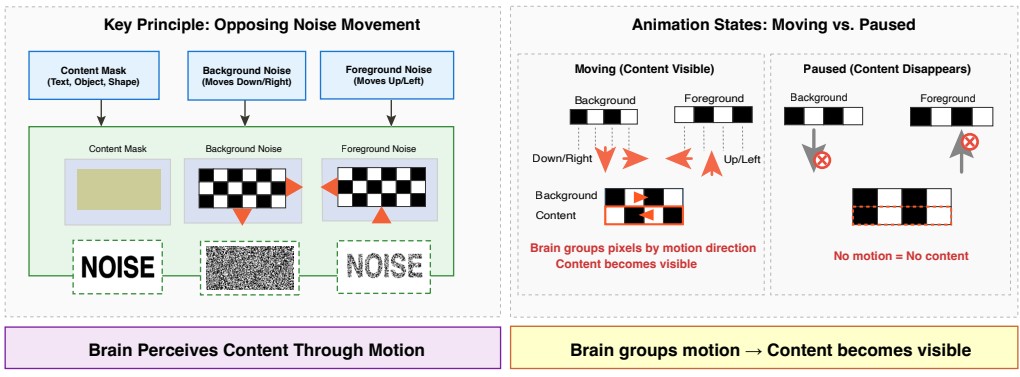

Figure 2: Illustration of the temporal encoding framework used in `SpookyBench`. **Left:** Core mechanism showing how content becomes visible through opposing motion patterns. A content mask defines regions where foreground noise (moving up/left) and background noise (moving down/right) are applied. When animated, the human visual system groups pixels with similar motion, causing the content to emerge. **Right:** Comparison between moving and paused states, demonstrating how content is only perceptible during animation and disappears when static, as individual frames contain only structured noise.

### 3.1 DATASET GENERATION

Figure 2 shows our proposed data generation framework. The dataset consists of specially designed videos that encode three types of content - words, images, and videos - using binary noise patterns with specific motion properties. In this approach, content is embedded within noise patterns such that individual frames appear as random noise, while the content becomes perceptible only when viewed as a temporal sequence. Our dataset encodes different types of content (Figure 3) through

temporal noise animations in the following categories: **1) Words:** Text rendered as masks in which the background noise and foreground noise move in opposite directions, making the text visible only through temporal movement. **2)Images:** Binary masks generated using SAM2 (Ravi et al., 2024) from single-object images generated using text-to-image model Flux (Labs, 2024), encoded using the same content mask animation approach as words. **3) Dynamic Scenes:** Depth maps extracted from videos in single-object tracking datasets LaSOT (Fan et al., 2019) and OTB2015 (Wu et al., 2015) using Video Depth Anything (Chen et al., 2025a). These are encoded using a technique in which pixels above a brightness threshold move while others remain static as shown in the algorithm 2.

## 3.2 TEMPORAL ENCODING FRAMEWORK

Our temporal encoding framework implements two distinct motion configurations as detailed in Algorithms 1 and 2. For words, and image masks (Algorithm 1), we employ opposing motion patterns between foreground and background. The content is first converted to a binary mask $M$ where $M(x, y) = 1$ represents foreground pixels and $M(x, y) = 0$ represents background. We generate two separate noise patterns $N_{bg}$ and $N_{fg}$ consisting of random binary values (0 or 255). During animation, foreground pixels sample from $N_{fg}$ with a positive offset that increases with time ($y + vt \mod h$), while background pixels sample from $N_{bg}$ with a negative offset ($y - vt \mod h$).

This creates the perception of opposing motion within and outside the masked regions. For video depth maps (Algorithm 2), we employ a threshold-based approach. Using depth maps $D$ extracted from videos, pixels with brightness values between lower

| Category | Basic SNR (dB) | Perceptual SNR | Temporal Coherence | Motion Contrast |
|---|---|---|---|---|
| Images | -46.95 ± 2.40 | -47.28 ± 2.28 | 8.00 ± 2.08 | 7.17 ± 5.00 |
| Dynamic Scenes | -48.95 ± 3.64 | -63.43 ± 5.74 | 21.91 ± 5.76 | -3.18 ± 10.17 |
| Text | -39.27 ± 1.58 | -49.18 ± 3.31 | 7.84 ± 0.65 | 8.26 ± 6.44 |

Table 2: Signal-to-Noise Ratio (SNR) metrics across Spooky-Bench categories.

and upper thresholds ($t_l \leq d \leq t_u$) are animated by sampling a noise pattern $N$ with a time-varying offset ($y + vt \mod h$), while pixels outside this range remain static. This creates the illusion that brighter regions (typically foreground objects) are moving while darker regions (typically background) remain static. The noise patterns are generated using binary values (0 or 255) in square blocks of variable size. We used different speckle sizes ranging from $1 \times 1$ to $3 \times 3$ pixels to investigate the effect of noise granularity on perception. For each speckle size, we also varied the noise density - the probability that a block is white versus black - using values of 10%, 30%, 50%, and 90%. These noise patterns arranged in pixel blocks create optimal perceptual conditions for human viewers while remaining challenging for vision language models. To ensure seamless animation, the noise patterns are made tileable by copying edge pixels to the opposite boundaries. All videos maintain consistent technical specifications: $960 \times 540$ pixel resolution, with an average duration of 7.11 seconds (ranging from 1.0 to 35.0 seconds) and an average of 333.5 frames per video. Text videos have a consistent duration of around 4 seconds; however, videos of dynamic scenes are longer, ranging up to 35 seconds. Figure 2 illustrates the structure of the data set and the encoding patterns in categories. We used binary masks for the images using SAM2 (Ravi et al., 2024). For videos, depth maps are extracted using Depth Anything V2 (Yang et al., 2025a) and Video Depth Anything (Chen et al., 2025a) from the LaSOT (Fan et al., 2019) and OTB2015 (Wu et al., 2015) datasets.

## 3.3 DATA STATISTICS

SpookyBench comprises 451 videos in three distinct categories, each requiring purely temporal reasoning for content identification. The dataset is distributed as follows: Text (46.6%, 210 videos), Object Images (40.8%, 184 videos) and Dynamic Scenes (12.6%, 57 videos). This distribution ensures comprehensive coverage of different temporal perception challenges while maintaining a natural frequency distribution that reflects real-world scenarios. Additionally, more dataset can be generated indefinitely through the data generator on our project page, thus the dataset size is essentially unlimited. The "Text" category contains common English words rendered through temporal noise patterns, enabling evaluation of models' ability to identify linguistic content through purely temporal cues. The "Object Images" category presents single objects extracted from high-quality images using segmentation techniques (Ravi et al., 2024), encoded with the same temporal animation approach. It also contains a synthetic silhouette of simple objects generated using DALL-E 3 (Betker et al., 2023) and flux (Labs, 2024).

### 3.3.1 ANALYSIS OF TEMPORAL METRICS

To ensure a rigorous quantification of the temporal information present in each video, we analyzed five key 2. SNR metrics that capture different aspects of the complexity and perceptibility of temporal patterns in SpookyBench, as shown in Table. These metrics provide insight into why temporal patterns might be visible to humans but challenging to detect by computational models.

**Basic SNR** measures signal-to-noise ratio in decibels:

$$\text{SNR}_B = 10 \log_{10} \left( \frac{P_S}{P_N} \right) \qquad (1)$$

where $P_S = \mathbb{E}[\|\nabla \mathbf{F}\|^2]$ is motion boundary energy derived from spatial gradients of optical flow field $\mathbf{F}(x, y) = (F_x, F_y)$, and $P_N = \text{Var}(I_0)$ is variance of the static frame $I_0$.

**Perceptual SNR** incorporates frequency-dependent visual sensitivity:

$$\text{SNR}_P = 10 \log_{10} \left( \frac{\|\mathcal{H}(B) \odot W\|^2}{\|\mathcal{H}(N) \odot W\|^2} \right) \qquad (2)$$

where $B$ is the average motion boundary strength, $N$ is the static noise frame, $\mathcal{H}$ is the 2D Fourier transform, $\odot$ denotes element-wise multiplication, and $W(f) = f \cdot e^{-f/f_0}$ is the contrast sensitivity weighting function with peak $f_0 \approx 0.1$ cycles/pixel.

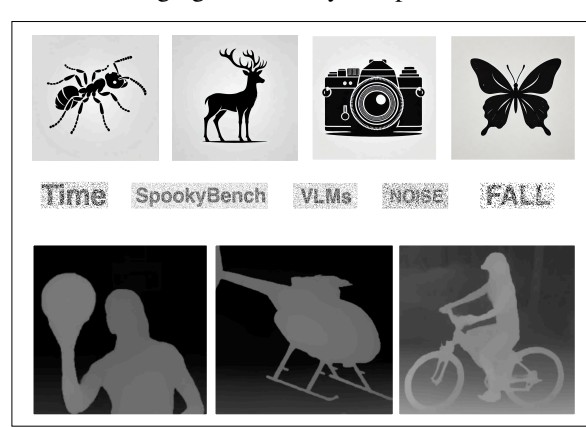

Figure 3: Noise generation process: (Top) masks applied for dynamic noise video generation, (Mid) word-specific mask, and (Bottom) depth map of video frame used for constructing noise-overlaid stimulus.

**Temporal Coherence SNR** quantifies motion consistency:

$$\text{SNR}_T = 10 \log_{10} \left( \frac{\text{Var}(C)}{\mathbb{E}[\text{Var}_{\text{local}}(C)]} \right) \qquad (3)$$

where $C = e^{-\text{Var}_\theta(\mathbf{F})} \cdot \mathbb{1}(\|\mathbf{F}\| > \tau)$ is the directional coherence map, $\text{Var}_\theta$ computes circular variance of flow direction angles over time, $\mathbb{1}$ is indicator function, $\tau$ is magnitude threshold, and $\text{Var}_{\text{local}}$ computes variance over small spatial neighborhoods.

**Motion Contrast SNR** measures foreground-background motion differentiation:

$$\text{SNR}_M = 10 \log_{10} \left( \frac{\|\boldsymbol{\mu}_M - \boldsymbol{\mu}_B\|^2}{\frac{1}{2}(\sigma_M^2 + \sigma_B^2)} \right) \qquad (4)$$

where $\boldsymbol{\mu}_M = \mathbb{E}[\mathbf{F} \mid M]$ and $\boldsymbol{\mu}_B = \mathbb{E}[\mathbf{F} \mid \neg M]$ are mean flow vectors within mask region $M$ and background region $\neg M$ respectively, $\sigma_M^2 = \mathbb{E}[\|\mathbf{F} - \boldsymbol{\mu}_M\|^2 \mid M]$ and $\sigma_B^2 = \mathbb{E}[\|\mathbf{F} - \boldsymbol{\mu}_B\|^2 \mid \neg M]$ are corresponding motion variances. The mask $M$ is estimated from the motion boundaries.

The distribution of these metrics reveals why current vision models struggle with **SpookyBench**: they lack mechanisms to leverage temporal coherence (particularly high in Dynamic Scenes, 21.91 ± 5.76 dB) and motion contrast (negative for Dynamic Scenes, -2.20 and -3.18 dB), while text stimuli benefit from higher basic SNR (-39.27 ± 1.58 dB), explaining the observed performance gap.

### 3.3.2 BINARY SNR THRESHOLD EFFECT IN DETECTION

Our analysis revealed a critical binary threshold phenomenon in detecting text within dynamic noise videos. The words exhibited negligible detection (∼0%) below 2.5dB SNR, but jumped to 85.7% accuracy above this threshold, displaying an abrupt rather than gradual transition as show in 4. Prompts performed best (40% accuracy), with Chain-of-Thought reasoning improving general identification tasks compared to direct prompting. This phenomenon parallels medical imaging diagnostics, where pathologies like microcalcifications in mammography become either entirely

**Algorithm 1** Content Mask Animation

1: **Input:** Content mask $M$, velocity $v$
2: **Output:** Animated frame $F_t$
3: Generate noise patterns $N_{bg}, N_{fg}$
4: **for** each pixel $(x, y)$ **do**
       ▷ Check pixel's mask status
5:  **if** $M(x, y)$ **then**
6:    $F_t(x, y) \leftarrow N_{fg}(x, y + vt \bmod h)$
         ▷ Foreground
7:  **else**
8:    $F_t(x, y) \leftarrow N_{bg}(x, y - vt \bmod h)$
         ▷ Background
9:  **end if**
10: **end for**

**Algorithm 2** Video Depth Map Animation

1: **Input:** Depth map $D$, thresholds $(t_l, t_u)$, velocity $v$
2: **Output:** Animated frame $F_t$
3: Generate noise pattern $N$
4: **for** each pixel $(x, y)$ **do**
5:  $d \leftarrow$ brightness from $D(x, y)$
6:  **if** $t_l \leq d \leq t_u$ **then**
7:    $F_t(x, y) \leftarrow N(x, y + vt \bmod h)$
         ▷ Moving noise
8:  **else**
9:    $F_t(x, y) \leftarrow N(x, y)$ ▷ Static noise
10:  **end if**
11: **end for**

visible or invisible based on specific SNR thresholds. The implications are significant: unlike perceptual phenomena that degrade gradually with noise, text detection functions as a step function, creating vulnerabilities in safety-critical applications. Just as radiologists cannot diagnose what remains invisible, language models cannot identify text below certain noise thresholds, leading to false certainties and potential catastrophic performance drops with minimal noise increases. This characteristic creates particular concerns for autonomous vehicles reading road signs or medical systems interpreting labels, while also exposing systems to adversarial attacks where slight SNR manipulations could render text completely undetectable.

## 4 EXPERIMENTS

### 4.1 EXPERIMENTAL SETUP

**Models.** We evaluated `SpookyBench` on both open source models (Video-LLaVA (Zhang et al., 2024b), LLaVA-NeXT-Video (Li et al., 2024c), TimeChat (Ren et al., 2024), InternVL2 (Chen et al., 2024f), Qwen2-VL (Wang et al., 2024b), Qwen2.5-VL (Bai et al., 2025) etc.) and closed source models (GPT-4o (Hurst et al., 2024), Gemini 2.0 Flash (DeepMind, 2025), and Gemini 1.5 Pro (Team et al., 2024). We design different prompts for each category. All the prompts are included in the Appendix C. All prompts instruct models to respond with only 1-5 words identifying the content. We input sequences of multiple video frames simultaneously for models that do not directly support video input.

**Setup.** We evaluate model performance using exact match accuracy between model responses and our labels. For the Text categories, each video has a single correct label $y_i$. For *Object Images and Dynamic Scenes* categories, we define a set of acceptable labels $Y_i = \{y_{i1}, y_{i2}, \ldots, y_{in}\}$ to account for semantic ambiguity. For example, a video showing "a man playing basketball" accepts responses such as "playing basketball," "man", "human", or "woman playing basketball" as correct. Formally, for each video $i$, given a model response $r_i$ and corresponding label or set of labels $L_i$ (where $L_i = y_i$ for Text or $L_i = Y_i$ for objects and dynamic scenes), we calculate the accuracy as: Accuracy $= \frac{1}{N} \sum_{i=1}^{N} \mathbb{1}(r_i \in L_i)$, where $\mathbb{1}$ is the indicator function that equals 1 if $r_i \in L_i$ and 0 otherwise, and $N$ is the total number of videos in the evaluation set. Despite this flexible evaluation protocol that accepts multiple valid responses for certain categories, none of the models tested produced responses that matched any of the acceptable options.

### 4.2 HUMAN EVALUATION

To evaluate human performance against our benchmark, we designed and conducted a controlled experiment involving human participants. We recruited a total of six human participants for this study, each independently evaluating all videos. Participants were instructed to view each video carefully and subsequently record their responses on an anonymized website in the following structured form: 1) **Perceptibility Rating (1-5):** Participants rated how perceptible the

presented word, shape, or object was, ranging from 1 (very difficult to perceive) to 5 (very clearly perceptible). This measure provided insights into the clarity and ease of visual grouping.

2) **Words/Objects Identification:** Participants typed out exactly what they identified in the video. This response directly tested the accuracy of their visual perception. We collect and evaluate participant responses using exact match criteria based on our predefined labels. Similar to the evaluation accuracy of the video language models for the categories of Object Images and Dynamic Scenes, we ac-

| Annotator | Text | | Images | | Dynamic Scenes | |
|---|---|---|---|---|---|---|
| | Acc(%) | Perc(1-5) | Acc(%) | Perc(1-5) | Acc(%) | Perc(1-5) |
| Annotator 1 | 99.5 | 4.7 | 99.5 | 4.7 | 96.5 | 4.3 |
| Annotator 2 | 98.6 | 4.8 | 98.4 | 4.9 | 91.2 | 4.0 |
| Annotator 3 | 99.5 | 4.9 | 97.2 | 4.5 | 94.7 | 4.4 |
| Annotator 4 | 97.6 | 4.6 | 96.7 | 4.5 | 91.2 | 4.0 |
| Annotator 5 | 100.0 | 4.8 | 99.5 | 4.7 | 99.0 | 4.7 |
| Annotator 6 | 98.0 | 4.7 | 97.8 | 4.5 | 93.0 | 4.2 |
| **Mean** | **98.9±0.7** | **4.8±0.0** | **98.2±1.1** | **4.7±0.1** | **94.3±3.1** | **4.3±0.1** |

Table 3: Human evaluation results showing accuracy and perceptibility ratings across different visual categories in `SpookyBench`.

cepted multiple correct responses to avoid ambiguity. Table 3 shows the average precision and the perception rating of different annotators for different categories. The results show high human performance across all categories: participants correctly identified Words with 98% accuracy, while Object Images had 92% accuracy. We also observe a very high perceptibility rating (4.8 for texts and 4.3 and 4.0 for Object images and Dynamic scenes, respectively) across all three categories. This shows that the human brain can easily extract coherent information in videos, which seems to be very difficult for video language models.

### 4.3 IMPACT OF FRAME RATES ON HUMAN AND MODEL ACCURACY

To examine whether temporal sampling affects performance, we evaluated both humans and VLMs across frame rates from 1 to 30 FPS. Three human participants tested 120 randomly sampled videos (40 per category) at 1, 5, 10, 20, and 30 FPS, while four VLMs (Qwen2-VL-7B, Qwen2.5-VL-7B, Qwen2.5-VL-3B, and GPT-4o) were evaluated using identical temporal downsampling. As shown in Tables 4 and 5, human accuracy remains above 95% at 20-30 FPS, degrades to 59.4% at 10 FPS, and drops to 0% at 1 FPS. In contrast, all VLMs achieved 0% accuracy across all frame rates. This demonstrates that temporal sampling frequency does not explain

| Category | 1 FPS | 5 FPS | 10 FPS | 20 FPS | 30 FPS |
|---|---|---|---|---|---|
| Images | 0.0 | 12.5 | 80.0 | 95.8 | 97.5 |
| Words | 0.0 | 10.8 | 35.8 | 95.8 | 95.8 |
| Videos | 0.0 | 15.0 | 62.5 | 93.3 | 93.3 |
| **Average** | **0.0** | **12.8** | **59.4** | **95.0** | **95.6** |

Table 4: Human accuracy (%) across different content categories at varying frame rates. Results are averaged across 3 participants on 120 videos (40 per category).

the performance gap between humans and current video-language models, indicating that VLMs lack the architectural mechanisms to process information conveyed through temporal patterns regardless of temporal resolution.

### 4.4 IMPACT OF FINETUNING ON MODEL ACCURACY

To investigate whether the performance gap stems from out-of-distribution data rather than architectural limitations, we finetuned two state-of-the-art video-language models on SpookyBench: InternVL2.5-8B and Qwen2-VL-7B. Both models were trained on 400 SpookyBench videos for 10 epochs using LlamaFactory (Zheng et al., 2024). Despite this targeted training on the exact task and data distribution, both models maintained 0% accuracy on the test set. This result demonstrates that the failure to decode temporal patterns is not attributable to domain mismatch or insufficient exposure to the task, but rather indicates a fundamental architectural inability to process information conveyed purely through motion without relying on spatial content.

## 5 RESULTS AND DISCUSSION

Table 1 presents the accuracy scores on the `SpookyBench` benchmark. Human participants achieved 98% accuracy under all test conditions. In contrast, all Video-VLMs scored 0% regardless of the type, size, or origin of the model. This pattern was held across all three task categories in our benchmark: temporal symbol recognition, temporal sequence understanding, and temporal pattern reasoning. We tested two different prompting strategies to determine if performance limitations could be overcome through interface modifications. First, we used direct prompts with basic instructions asking the models to identify content in the videos. Next, we implemented

chain-of-thought prompts with explicit guidance to focus on temporal patterns rather than individual frames. As shown in Table 1, none of these approaches yielded improvements. All models maintained 0% accuracy across all prompting conditions, suggesting that the limitation is inherent in the model architectures rather than a matter of optimization or prompt design. Examination of model output revealed consistent failure modes when processing **SpookyBench** videos.

Across all models tested, we observed attempts to extract information from individual frames rather than temporal patterns. When ex-

| Model | Qwen2-VL-7B | Qwen2.5-VL-7B | Qwen2.5-VL-3B | GPT-4o |
|---|---|---|---|---|
| **Accuracy (%)** | 0.0 | 0.0 | 0.0 | 0.0 |

Table 5: VLM accuracy (%) averaged across all tested frame rates (1-30 FPS).

plicitly prompted to consider temporal changes, the models acknowledged the instruction but still failed to identify the patterns. Fine-tuned models produced outputs that mimicked training examples without correctly identifying test patterns. In particular, specialized temporal models like TimeChat (Ren et al., 2024), which were specifically designed for fine-grained temporal understanding, failed at the same rate as general-purpose models. This suggests that the limitation extends beyond general Video-VLMs to models explicitly optimized for temporal tasks.

**Architectural Implications for Vision Models.** Distinctive signal profiles in **SpookyBench** demonstrate a fundamental gap between human and machine perception of temporal information. Current vision models struggle with **SpookyBench** stimuli primarily because they: (1) lack robust temporal integration mechanisms that could leverage high temporal coherence, (2) process information primarily through spatial rather than temporal channels, and (3) fail to perform motion-based figure-ground segregation effectively.

The consistently high temporal coherence values in Dynamic Scenes, coupled with their poor static-frame metrics, suggest that successful models must implement recurrent processing or attention mechanisms that operate across extended temporal windows rather than focusing on frame-level feature extraction. The negative motion contrast observed in Dynamic Scenes further indicates that models require more sophisticated motion segregation capabilities to match human perceptual abilities in dynamic visual environments. These findings highlight the need for architectural innovations that specifically address temporal processing limitations. Future models should incorporate dedicated

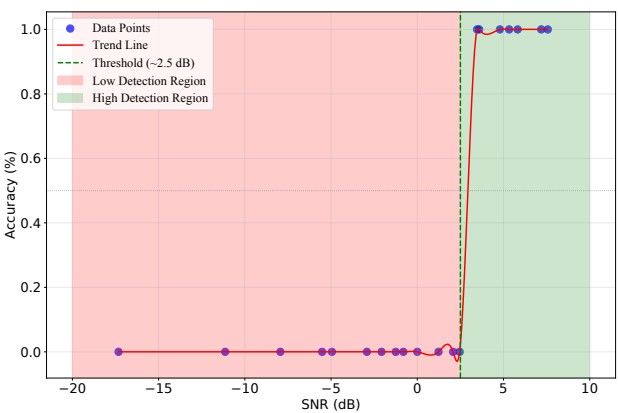

Figure 4: Analysis of effects of SNR on detecting words with direct prompting and chain of thought prompting.

temporal coherence pathways, motion contrast analysis, and longer temporal integration windows to bridge the perception gap demonstrated by **SpookyBench**.

## 6 CONCLUSION

In this paper, we introduced **SpookyBench**, a novel benchmark designed to evaluate the temporal reasoning capabilities of video-language models by isolating temporal understanding from spatial comprehension. Our experiments revealed a striking performance gap: while humans effortlessly achieve 98% accuracy on tasks requiring pure temporal pattern recognition, all tested models, including state-of-the-art open and closed-source systems, fail completely with 0% accuracy. This consistent failure across different model architectures, scales, and prompting strategies highlights a fundamental limitation in current video understanding approaches, which typically process spatial features first and then establish temporal connections, rather than integrating spatio-temporal information simultaneously. The benchmark effectively exposes the *time blindness* of current architectures that remain hidden in conventional evaluation settings where spatial features can provide shortcuts to correct answers. We hope that **SpookyBench** will inspire the development of next-generation temporal-connected models.

ETHICS STATEMENT

This work introduces SpookyBench, a synthetic benchmark for evaluating temporal understanding in video-language models, and does not involve collection of personally identifiable information or creation of harmful content. The human evaluation component involved six volunteer participants who provided informed consent and were free to withdraw at any time. All participant responses were anonymized and stored securely. We used publicly available datasets (LaSOT, OTB2015) and models under their original licenses, and evaluated both open-source and commercial VLMs following their respective usage policies. While exposing fundamental limitations in current video-language models may impact their deployment in safety-critical applications, we believe this transparency is essential for responsible AI development. The synthetic nature of our dataset eliminates concerns about data consent or privacy violations. We acknowledge that improved temporal understanding capabilities could potentially be misused, but the same capabilities are fundamental for beneficial applications in medical imaging, autonomous systems, and accessibility technologies. We encourage responsible development and deployment practices, including human oversight in critical applications and adherence to existing AI safety guidelines.

REPRODUCIBILITY STATEMENT

SpookyBench is generated using fully deterministic algorithms detailed in Algorithms 1 and 2, with specific parameters for noise generation, motion patterns, and video specifications clearly documented. We will release: (i) complete code for dataset generation with all hyperparameters (velocity values, noise densities, speckle sizes, threshold ranges); (ii) the full SpookyBench dataset with 451 videos across three categories; (iii) exact evaluation prompts for both direct and chain-of-thought strategies; (iv) model evaluation scripts with specific version numbers and inference parameters; and (v) fine-tuning configurations used with LlamaFactory. All SNR metrics are computed using the mathematical formulations provided in Section 3.3. We document the exact model versions evaluated (e.g., Qwen2.5-VL-7B-Instruct, InternVL2.5-8B) and will provide environment specifications including framework versions, hardware details, and computational requirements. The human evaluation methodology, including participant instructions and response collection protocols, is fully documented to enable replication of the human baseline results.

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

# APPENDIX

## A  USE OF LARGE LANGUAGE MODELS

An LLM was used to help solely polish the writing of the paper, while all methods, ideas and experiments were prepared and carried out entirely by the authors.

## B  DATA STATISTICS

Figure 5 shows the data distribution explained in the section 3.3.

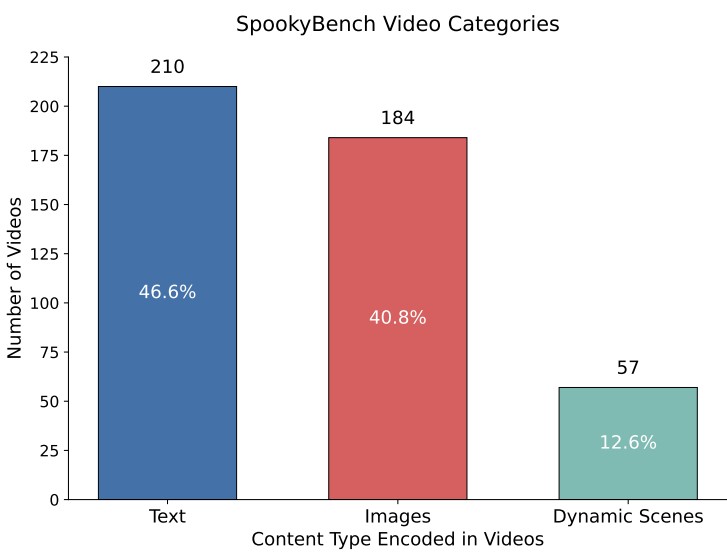

Figure 5: Distribution of the **SpookyBench** dataset across three video categories. Each category represents a different type of content encoded through temporal noise patterns: *Text*, *Object Images*, and *Dynamic Scenes*.

## C  PROMPT DESIGN FOR EVALUATION

Prompt design significantly affects the performance of vision-language models (Jin et al., 2021; Gu et al., 2023). Considering this fact, we performed careful prompt engineering to ensure fair and comprehensive evaluation. We developed a systematic prompting methodology that builds on established principles while introducing novel elements specific to temporal pattern recognition.

### C.1  PROMPT DESIGN PRINCIPLES

We designed our prompts based on three key principles:

1. **Specificity**: Each prompt explicitly states that the content is encoded through temporal patterns to direct attention to motion-based cues rather than static frame analysis.

2. **Category targeting**: We created specialized prompts for each content category (text, objects, dynamic scenes) to account for the different perceptual mechanisms involved in each.

3. **Constrained response format**: All prompts request brief, specific answers (1-3 words) to ensure objective evaluation and minimize the influence of language generation capabilities.

**Text-Specific Direct Prompt**

This video contains text encoded through temporal patterns. What specific word or phrase do you see? The text is only visible through the temporal changes in the video. Please respond with just the text you identify.

**Object-Specific Direct Prompt**

This video contains a common object encoded through temporal patterns. Individual frames may appear as noise, but an object is visible through the temporal changes. What object do you see? Please respond with just the object name.

**Dynamic Scenes-Specific Direct Prompt**

This video contains movement or action encoded through temporal patterns. The content is only visible through temporal changes, not in individual frames. What is being shown? Please respond with just 1-3 words describing what you see.

Figure 6: Category-specific direct prompts for the SpookyBench benchmark. These prompts test immediate pattern recognition without step-by-step guidance.

**Text-Specific CoT Prompt**

This video encodes text through temporal patterns. To identify it:

1. Look for areas where opposing motion patterns reveal letters
2. Focus on the overall word or phrase that emerges
3. Read the specific text content

Please respond with just the text you identify.

Figure 7: Category-specific chain-of-thought prompts for the SpookyBench benchmark. These prompts provide explicit step-by-step guidance to test structured temporal reasoning.

## C.2 DIRECT VS. CHAIN-OF-THOUGHT PROMPTING

We implemented two distinct prompting strategies to investigate different aspects of temporal understanding.

- **Direct prompts** test immediate pattern recognition without explicit guidance, similar to how humans naturally perceive temporal patterns without conscious step-by-step processing.
- **Chain-of-Thought (CoT) prompts** provide explicit steps to guide attention and processing, testing whether models could benefit from structured reasoning about temporal patterns.

The figures below present our category-specific prompts for both strategies, which were carefully optimized through pilot testing to maximize clarity while maintaining consistent evaluation criteria across categories.

## C.3 PROMPT EFFECTIVENESS ANALYSIS

Our experiments revealed that, surprisingly, neither prompt strategy improved model performance on the SpookyBench tasks. All tested models achieved 0% accuracy regardless of prompt type,

---

**Object-Specific CoT Prompt**

This video encodes an object through temporal patterns. To identify it:

1. Look for areas where motion patterns reveal object contours

2. Focus on the overall silhouette and form that emerges

3. Determine what specific object is represented

Please respond with just the object name.

---

**Dynamic Scenes-Specific CoT Prompt**

This video encodes movement through temporal patterns. To identify it:

1. Look for areas where temporal changes reveal motion

2. Focus on the action or activity that emerges from the pattern

3. Identify the specific movement or object in motion

Please respond with just 1-3 words describing what you see.

---

Figure 8: Additional category-specific chain-of-thought prompts for object images and dynamic scenes.

indicating a fundamental limitation in their ability to process purely temporal information rather than a prompt engineering issue.

The complete ineffectiveness of even carefully engineered prompts across all tested models further strengthens our argument that current video-language models lack the fundamental architectural mechanisms needed for processing purely temporal patterns.

# D    IMPACT OF FPS

To test the impact of frame rate on both human and VLM performance, we conducted additional experiments examining how temporal sampling affects the ability to perceive information encoded purely through motion patterns. This analysis addresses a critical question: could the performance gap between humans and VLMs be attributed to differences in temporal sampling rather than fundamental architectural limitations?

We evaluate both human participants and video-language models across multiple frame rates ranging from 1 to 30 FPS. For the human study, three participants from our human annotator were tested on 60 randomly sampled videos (15 from each category) at frame rates of 1, 5, 10, 20, and 30 FPS. For the VLM evaluation, we test four state-of-the-art models: Qwen2-VL-7B, Qwen2.5-VL-7B, Qwen2.5-VL-3B, and GPT-4o. We applied the same temporal downsampling approach, ensuring that models received the exact number of frames corresponding to each target frame rate.

Table 4 presents the human performance results across different frame rates and content categories. Human accuracy remains remarkably robust at higher frame rates, maintaining over 95% accuracy at 20-30 FPS across most categories. Performance begins to degrade at 10 FPS, dropping to 59.4% on average, with particularly pronounced effects on the Words category (35.8%). At extremely low frame rates (1-5 FPS), human performance drops substantially, reaching only 0% at 1 FPS.

In stark contrast, Table 5 shows that all tested VLMs achieved 0% accuracy regardless of frame rate. This consistent failure across the entire range of tested frame rates, from 1 FPS to 30 FPS, demonstrates that temporal sampling frequency is not the limiting factor for current video-language models.

These results reveal a fundamental difference in how humans and current VLMs process temporal information. While human performance degrades gracefully as frame rate decreases, particularly

| SNR Metric | Value (dB) |
|---|---|
| Basic SNR | -49.07 |
| Perceptual SNR | -55.02 |
| Temporal Coherence SNR | 7.18 |
| Motion Contrast SNR | 14.24 |
| Combined SNR | -20.61 |

Table 6: Signal-to-Noise Ratio Analysis for Ant Silhouette Video

below 10 FPS where temporal patterns become harder to perceive, VLMs show no improvement even at optimal frame rates. This finding effectively rules out temporal undersampling as an explanation for the observed performance gap. The graceful degradation in human performance at lower frame rates aligns with our understanding of human temporal perception, where motion detection requires sufficient temporal resolution. However, the complete insensitivity of VLMs to frame rate variations suggests that these models are not engaging with temporal information in any meaningful way, regardless of how much temporal data is provided.

This analysis strengthens our core argument that current video-language models lack the fundamental architectural mechanisms needed to process information conveyed purely through temporal patterns, independent of spatial content quality or temporal sampling considerations.

# E   TEMPORAL MOTION COHERENCE ANALYSIS

Visual content in noise presents a significant challenge for perception. However, temporal coherence and motion boundaries provide powerful cues that enable the human visual system to extract meaningful shapes even from extremely noisy stimuli. We present a comprehensive analysis of these phenomena using our SpookyBench dataset, specifically examining how temporal information facilitates shape perception in high-noise conditions.

## E.1   MOTION-BASED PERCEPTION IN NOISY ENVIRONMENTS

Our analysis demonstrates that even when individual frames contain low signal-to-noise ratios (SNR), temporal integration of motion information allows for robust shape perception. Figure 9 shows the motion direction coherence map of an ant silhouette, revealing how consistent motion patterns across frames enable object identification despite significant noise.

The importance of temporal integration for shape perception is further demonstrated in Figure 11, which shows both the average motion boundary strength and its overlay on a noisy frame. Motion boundaries emerge clearly despite the extreme noise levels in individual frames (measured at -49.07 dB basic SNR), highlighting the importance of temporal information for noisy visual content.

## E.2   SIGNAL-TO-NOISE RATIO ANALYSIS

To quantify the perceptual phenomenon observed in our stimuli, we conducted detailed SNR analysis. Table 6 presents the results for the ant silhouette video shown in the figures above. Notably, while the basic and perceptual SNR metrics show extremely low values (-49.07 dB and -55.02 dB respectively), the temporal coherence SNR and motion contrast SNR reveal significantly higher values (7.18 dB and 14.24 dB), demonstrating that temporal information provides crucial signal enhancement that supports human perception.

This analysis provides important insights into how the human visual system utilizes temporal information to extract meaningful content from extremely noisy visual stimuli. The stark contrast between the negative frame-based SNR values and the positive temporal SNR metrics directly supports our hypothesis that temporal integration plays a crucial role in human perception of noisy visual content. These findings align with the high human evaluation accuracy reported in the main paper, where participants achieved over 95% accuracy for Object Images despite the extreme noise levels in individual frames.

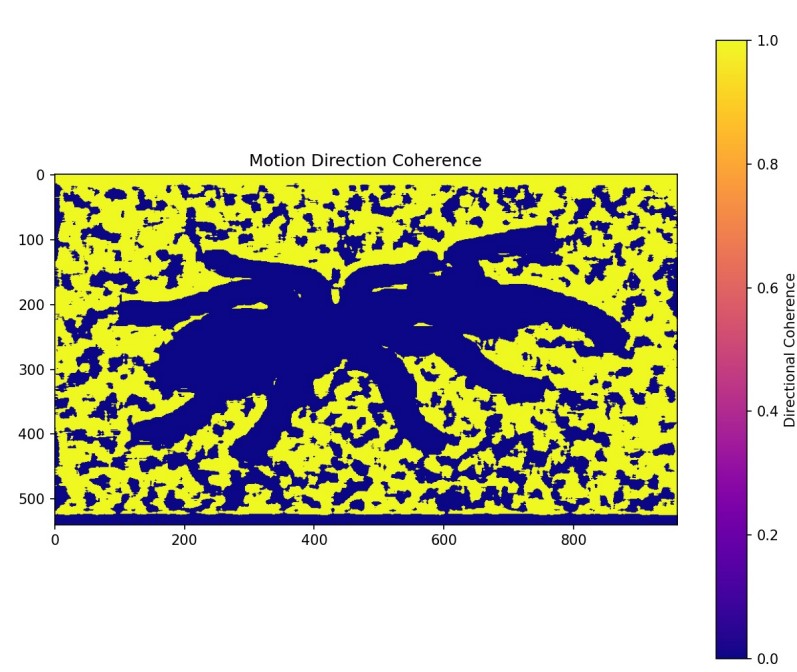

Figure 9: Motion Direction Coherence visualization for the ant silhouette video. Yellow regions (high coherence value of 1.0) indicate areas where motion direction remains consistent across frames, while blue regions (coherence value of 0.0) represent the silhouette itself.

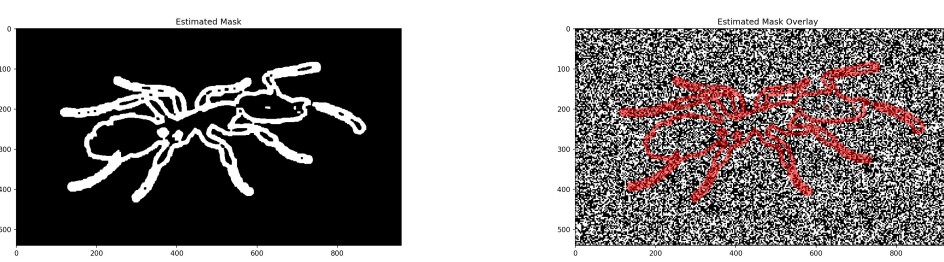

(a) Estimated object mask extracted from temporal motion coherence.

(b) Estimated mask overlay (red) on a single noise frame.

Figure 10: Shape extraction through temporal integration. These visualizations demonstrate how object shape can be recovered from noisy video sequences.

## F   ADDITIONAL IMAGES

We present additional images from the analysis of temporal motion coherence across all categories in SpookyBench. Figures 12, 13, 14, and 15 show motion boundaries, boundary overlays, estimated masks, and mask overlays for different examples in our dataset, demonstrating the varying effectiveness of temporal integration across different content types. In Figure 15, we observe that the

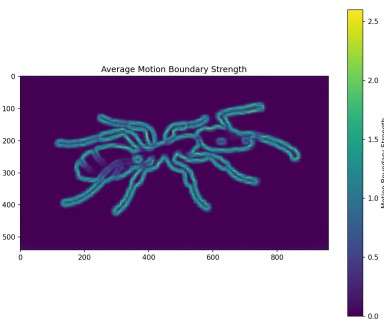
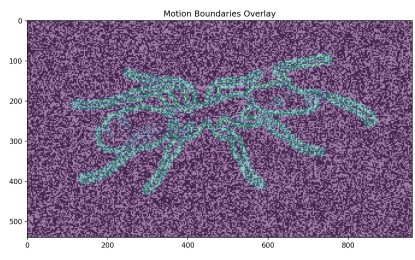

(a) Average motion boundary strength across frames.

(b) Motion boundaries (teal) overlaid on a single noise frame.

Figure 11: Motion boundary analysis demonstrating how temporal integration can extract object boundaries despite extremely noisy individual frames.

temporal motion coherence based method does not perform effectively for the case of videos. Since the Dynamic Scenes category contains real-life videos with complex motion patterns, several factors contribute to the reduced clarity observed in dynamic content:

1. **Distributed Motion Patterns**: Human movement involves multiple articulated body parts moving in different directions simultaneously, creating competing motion signals that fragment coherent boundaries.

2. **Non-rigid Deformation**: Dynamic content involves continuous shape changes throughout motion sequences, making consistent boundary extraction significantly more challenging than static objects.

3. **Complex Temporal Dynamics**: Mechanical motions in examples like Plane_2 and Bicycle_10 create temporal discontinuities that disrupt motion coherence essential for shape perception.

This comprehensive analysis demonstrates the systematic nature of temporal pattern recognition challenges across all SpookyBench categories, with each content type presenting distinct perceptual and computational difficulties that current video-language models fail to address.

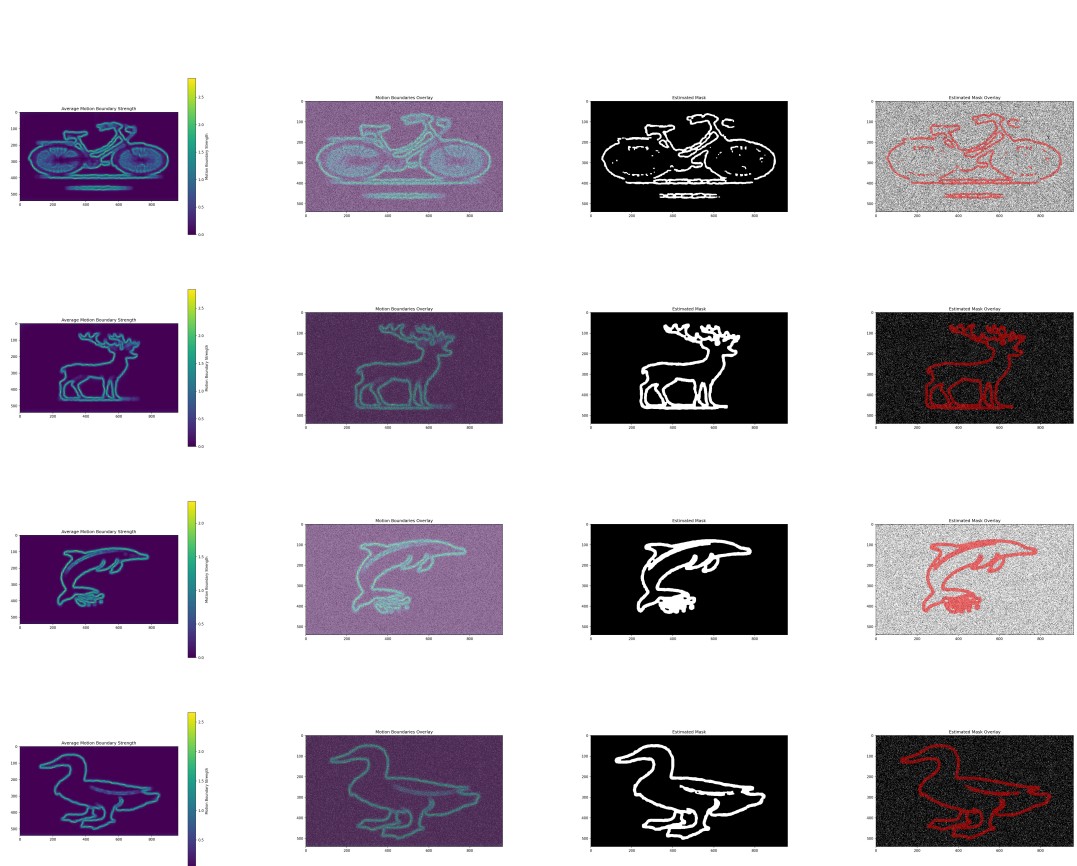

Figure 12: Temporal motion coherence analysis for Images category (Part 1). Each row shows motion boundaries, boundary overlay, estimated mask, and mask overlay for: Cycle, Deer, Dolphin, and Duck (top to bottom).

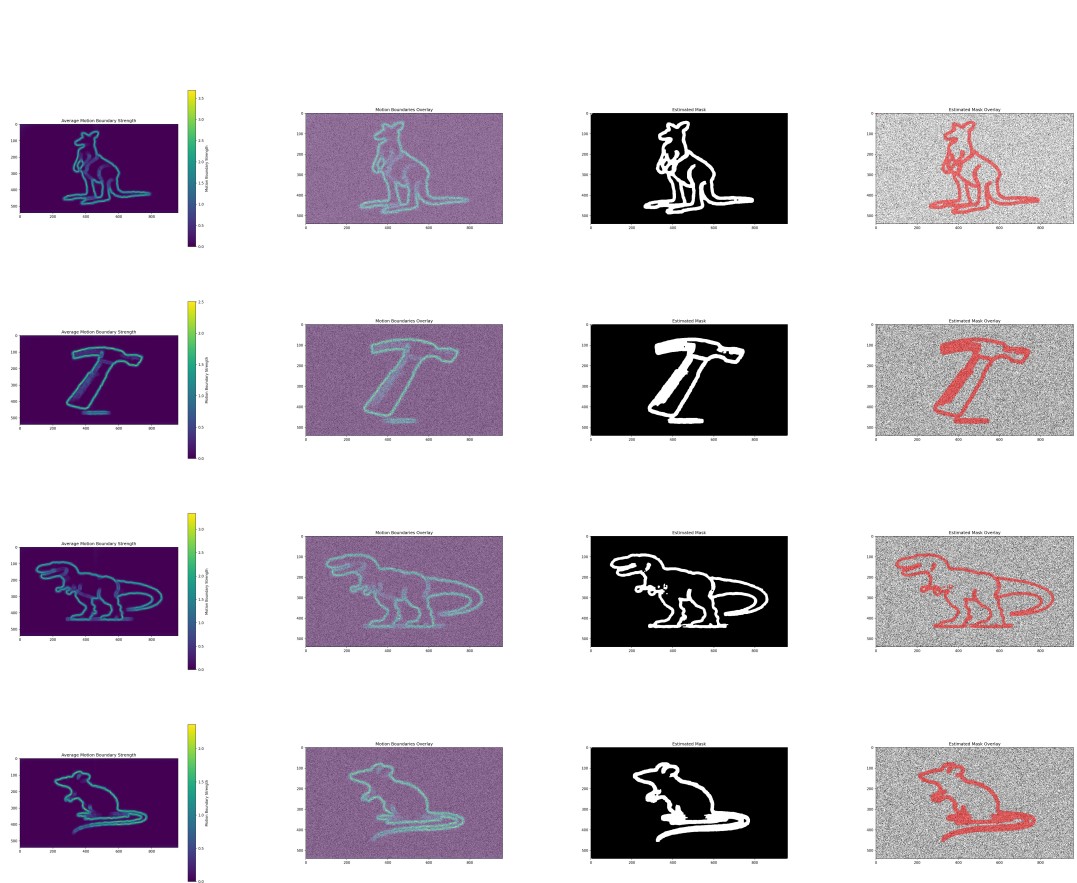

Figure 13: Temporal motion coherence analysis for Images category (Part 2). Each row shows motion boundaries, boundary overlay, estimated mask, and mask overlay for: Kangaroo, Hammer, T-Rex, and Mouse (top to bottom).

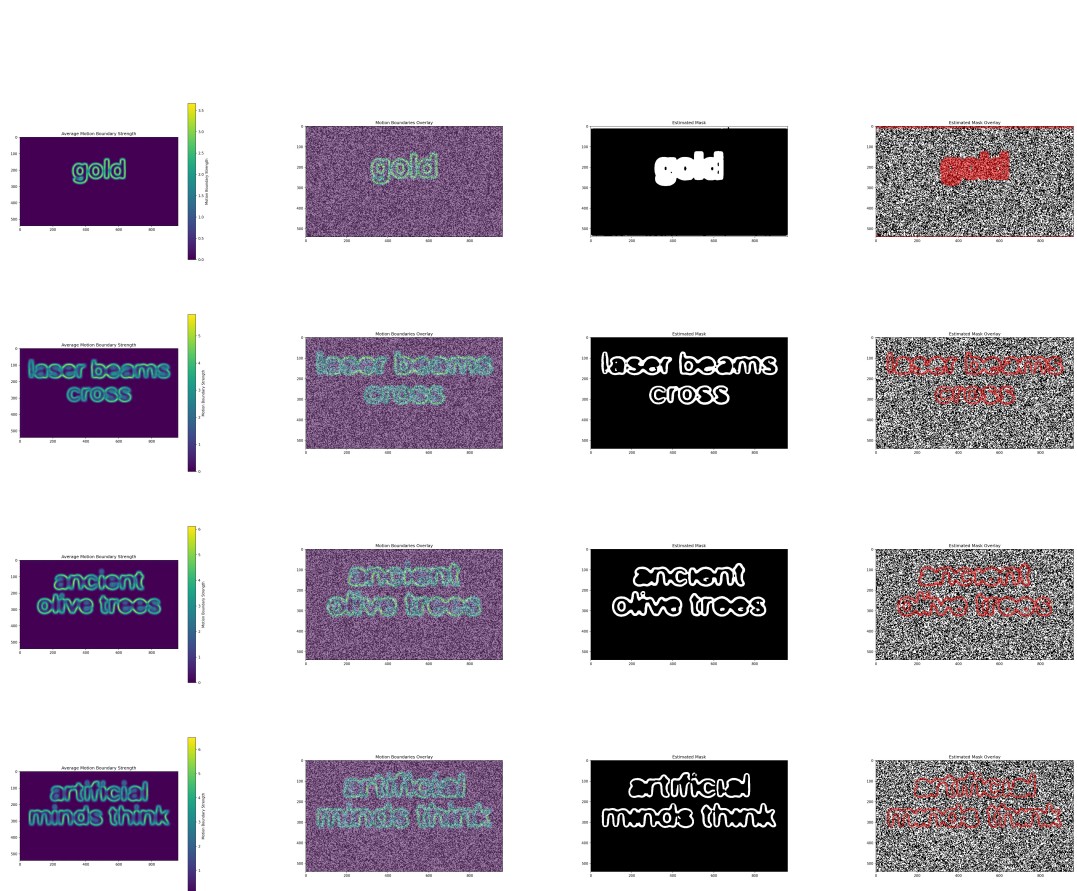

Figure 14: Temporal motion coherence analysis for Words category. Each row shows motion boundaries, boundary overlay, estimated mask, and mask overlay for: Gold, Laser Beams Cross, Ancient Olive Trees, and Artificial Minds Think (top to bottom).

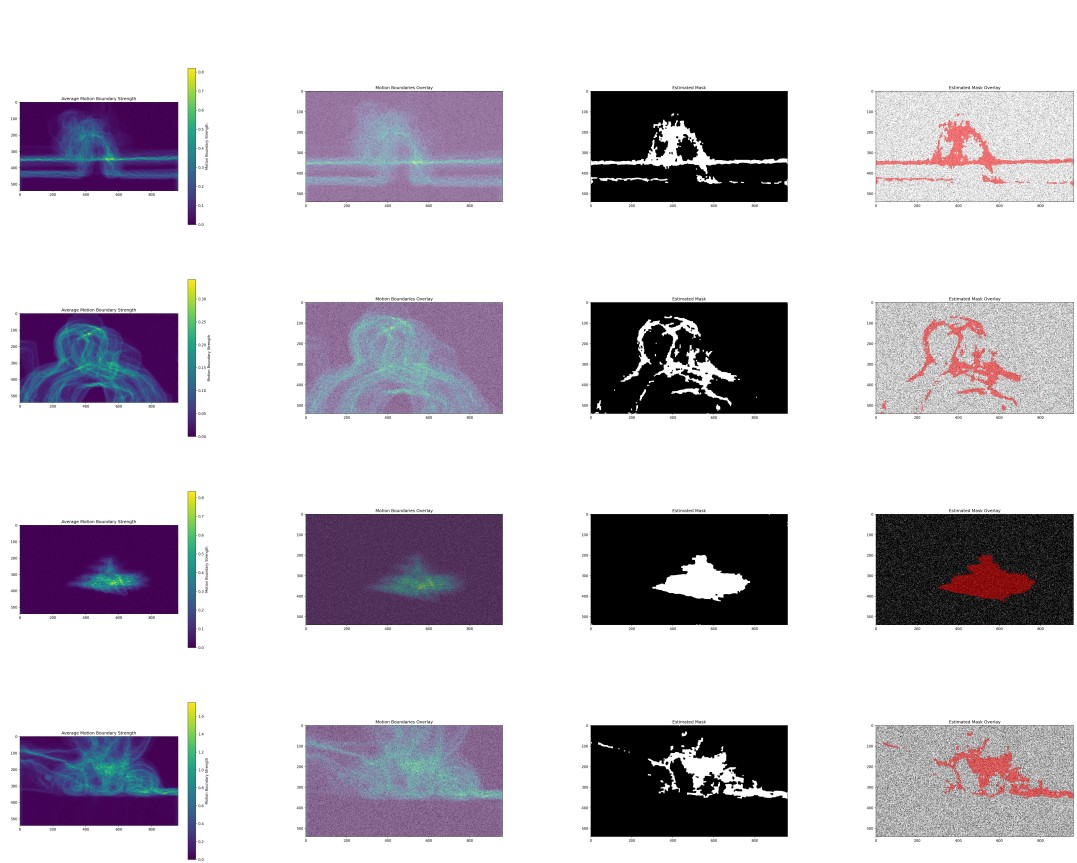

Figure 15: Temporal motion coherence analysis for Videos category. Each row shows motion boundaries, boundary overlay, estimated mask, and mask overlay for: Human_6, Man_1, Plane_2, and Bicycle_10 video sequence(top to bottom).

