# OpenReview forum: "Time Blindness: Why Video-Language Models Can’t See What Humans Can?"
_ICLR.cc/2026/Conference — ICLR 2026 Conference Withdrawn Submission_

### Official Review · Reviewer_byj5 · 2025-10-21

**Soundness:** 2
**Presentation:** 3
**Contribution:** 2
**Rating:** 2
**Confidence:** 3

**Summary:**

This paper presents SpookyBench, a synthetic video benchmark that aims to eliminate spatial cues from temporal reasoning tasks through presenting each frame as composed of only pure noise pixels. The content and background pixels are animated to move with different dynamic patterns to reveal the content of each video.

**Strengths:**

1. The idea of presenting each static frame as pure noise and only revealing the content through analysing video temporally is novel and interesting.
2. They authors have done extensive experiments on models across different families, architectures, and sizes to show that they all achieve zero performances.
3. Studies of 4 SNR metrics to provide insights on the temporal information in the videos.
4. The authors also included two separate experiments aiming to address the impact of frame rates and the impact of finetuning to support their claims.
5. The paper is clearly written and easy to follow.

**Weaknesses:**

1. Admittedly, there are different levels of spatial information that models can exploit in temporal reasoning tasks. However, I remain unconvinced about the claim that spatial features can be fully decoupled from temporal features when studying temporal reasoning in VLMs, including in this benchmark. Humans still need to identify patterns presented in this benchmark by diagnosing the spatial dynamics of the noise chunks/patches. It is essentially spatial information manifested in a different form.
2. All zero performance across all model families, architectures, sizes, and video frame rates are not so informative about models’ current true capabilities and ways to improve them. The experiment designs could be more fine-grained to differentiate models’ capabilities in understanding temporal patterns (see Questions for detail).
3. No ablation studies to gauge the effect of injected noise in the benchmark. While models can lack the ability to fully reason temporally, the bad performance could arise from the synthetically implanted noise. As current models are still prone to make mistakes with injected noise corruptions [1], if strong claims were to be made about models temporal reasoning capability, it is crucial to differentiate the effect of this capability from models’ noise resistance capability.
4. The authors claim that the dataset size is unlimited because of their data generator. However, due to the free-form QA design, it would require a lot of manual effort to check the answers against the groudtruth. It would also be hard to ensure that the predefined set of acceptable labels are exhaustive. For examples, there are many variations of the deer species whose names do not even contain the keyword "deer".
5. I am also concerned about the practical real-world application of this benchmark as the data is all synthetic and composed of noise. If model performance were to be improved through architectural modifications and/or training, the model's generalization ability to apply this type of temporal reasoning to real-world problems would still remain a question. \
  (a). Benchmarks composed of both natural and synthetic videos already exist [2] to expose models’ exploitation of static frame information in temporal reasoning. \
\
[1] Usama, Muhammad, et al. "Analysing the Robustness of Vision-Language-Models to Common Corruptions." arXiv preprint arXiv:2504.13690 (2025). \
[2] Shangguan, Ziyao, et al. "Tomato: Assessing visual temporal reasoning capabilities in multimodal foundation models." arXiv preprint arXiv:2410.23266 (2024).

**Questions:**

1. Are all zero performance at all useful for the current model development?\
  (a). Is it possible that some more capable models have recognized some temporal patterns in the video, however, as they are largely trained on natural videos such that even a static frame of just the object filled with noise in a clean background (or the reverse) would already pose challenges to the models in identifying the object?\
  (b). Have the authors considered testing with multiple-choice questions to try to differentiate between different models by looking at baselines like random choice, or frequent choice? The reasoning behind this is that some models might have detected some temporal patterns and only need a little bit additional information from the choices, whereas less capable models might perform even worse than random choice. Benchmark designs like this will better differentiate the models and help researchers gain more insight in how to improve them.
2. How can you make sure this is more than a data problem? I understand that authors have finetuned two models, 7B and 8B for 10 epochs on 400 datapoints. However, as the synthetic videos in this benchmark appear very out-of-distribution, it is unclear if this level of finetuning is sufficient to claim OOD not being a factor of all zero performances. Have the authors considered designing experiments that would help gauge how much the models’ failures are due to the videos being highly out-of-distribution, as opposed to a lack of temporal reasoning ability?
3. Have the authors considered using traditional computer vision models as experiment baselines, since solving the tasks depends on identifying the outlines/boundaries of object/text/dynamic scenes from the movements of the noise pixels? This might help provide insights on the applicability of the benchmark in model development.\
\
Minor:
1. As discussed in the paper and also shown in Figure 2, the animations are 2D. However, in Algorithm 1&2, the pixel manipulations seem to all be in the y- direction, which is only 1D. Is this expected?
2. The authors mentioned five key SNR metrics, but there are only four of them listed: Basic SNR, Perceptual SNR, Temporal Coherence SNR, and Motion Contrast SNR. Is this a typo?

---

### Official Review · Reviewer_Bw8j · 2025-10-26

**Soundness:** 2
**Presentation:** 2
**Contribution:** 1
**Rating:** 2
**Confidence:** 3

**Summary:**

The paper introduces SpookyBench, a benchmark to test whether video-language models can understand information encoded in temporal patterns rather than spatial features. Humans easily perceive these patterns, but all reported models fail completely.

**Strengths:**

- Originality: For benchmarks, the idea of removing all spatial cues to isolate temporal understanding is interesting.

- Quality: The authors evaluated both open and closed-source models to support the claims.

- Clarity: The writing is organized with clean figures.

- Significance: The paper challenge the assumption in video-language modeling that temporal reasoning can be treated as secondary to spatial processing, showing that current models fail completely when information exists only in time.

**Weaknesses:**

- The paper reports 0% for every model seems intended to grab attention, but it lacks rigor. Also, the results table reports zero accuracy, but lacks qualitative error analysis.
- Chain-of-thought vs. direct prompting yielded 0 % accuracy, but analysis of intermediate reasoning traces is missing (Table 1).
- The comparison is just binary, “pure noise” vs. “visible images”. The authors may add intermediate variants with partial spatial cues to measure a continuous degradation curve. Ablations are needed here.
- The paper fails to propose or evaluate any solution or mitigation strategy, making it difficult to assess its practical contribution beyond exposing an existing limitation. The work would be significantly strengthened by exploring even preliminary directions for addressing this limitation.
- There are not enough direct comparisons with other temporal-related benchmarks. While the authors cite a few in the related work, a more thorough discussion and direct comparison are needed. Including a summary table that highlights how SpookyBench differs from or complements existing datasets would strengthen the paper.

**Questions:**

- The paper draws inspiration from human perception. Can authors link this more explicitly to known neuroscientific mechanisms?
- Are input configurations across models fair and comparable? Please provide detailed information about the number of frames, sampling intervals, and preprocessing used for each model.
- Why does fine-tuning still yield 0% accuracy? Can authors clarify more experimental details?

**Details Of Ethics Concerns:**

No ethics concerns.

---

### Official Review · Reviewer_V4XF · 2025-11-01

**Soundness:** 3
**Presentation:** 3
**Contribution:** 4
**Rating:** 8
**Confidence:** 5

**Summary:**

This paper introduces a new benchmark, SpookyBench, which aims to isolate pure temporal understanding by presenting information only through temporal patters and individual frames resemble noise.

All meaningful information is encoded exclusively in the temporal domain through dynamic patterns of texts, images and video depth maps, while individual frames contain only structured noise. The background noise moves in certain directions while the foreground mask moves in opposite direction causing the content to emerge. In total, it has 451 videos.

The authors evaluate a number of closed and open models and find none of them are able to get past 0% accuracy while humans achieve near perfect accuracy. Furthermore, they also finetune two VLMs on such videos and yet see no gain in performance.

**Strengths:**

1.  In an era of saturating benchmarks, it is refreshing to see a benchmark that shows clear separation between human and machine ability. While there is a lot of work on temporal understanding focused on language [e.g., 1, 2], this work evaluates purely temporal understanding with a creative benchmark.

2. The benchmark construction is very thorough and mathematically well-explained.

3. The central finding is striking that none of the VLMs get past 0% accuracy even with fine-tuning. This really shows that the problem is right at the level of tokenising frames independently already loses critical temporality in the signal.

4. The paper is well written.

[1] TVBench: Redesigning Video-Language Evaluation. Cores et al.
[2] Enhancing Audio-Language Models through Self-Supervised Post-Training with Text-Audio Pairs. Sinha et al.

**Weaknesses:**

1. I think the claim that "overcoming this benchmark will need new architectures" is not substantiated. For example, if one enables tool use to use Gaussian Blur or use temporal frame difference, would that solve the task? While it is unfair to provide additional tools, I am not sure if humans also implicitly use some sort of preprocessing to only focus on temporal signals.

2. Perhaps, some more visual prompting baselines could be included. For example, if one augments the frames with temporal-difference frames, does it improve performance?

**Questions:**

Most of these are just based on the weaknesses I stated.

1. If one enables tool use to use Gaussian Blur or use temporal frame difference, would that solve the task?

2. If one augments the frames with temporal-difference frames, does it improve performance?

3. Do you have thoughts on tokenisation and why that may be the source of problem for a task like this? And possible solutions?

---

### Official Review · Reviewer_uxDn · 2025-11-02

**Soundness:** 2
**Presentation:** 2
**Contribution:** 2
**Rating:** 2
**Confidence:** 4

**Summary:**

The paper introduces SpookyBench, a benchmark for evaluating temporal visual understanding in video-language models using temporal noise patterns where content is only visible through motion (based on random dot motion techniques from psychophysics). The benchmark comprises 451 videos across three categories (text, images, dynamic scenes) where individual frames appear as random noise but reveal content when viewed temporally. Experiments show humans achieve 98% accuracy while all tested VLMs (including commercial ones like GPT-4o and Gemini) achieve 0% accuracy across different prompting strategies and frame rates, revealing a fundamental gap in temporal processing capabilities.

**Strengths:**

The benchmark effectively exposes critical limitations and differences in current video-language models' temporal reasoning capabilities and provides a valuable diagnostic tool for the community. The human-machine performance gap (98% vs 0%) highlights an important architectural deficiency that remains hidden in conventional evaluations where spatial features provide shortcuts.

**Weaknesses:**

-- limited technical novelty: The temporal noise patterns are established techniques from psychophysics (random dot motion), the results are somewhat expected given models' known reliance on spatial features, and the pattern generation algorithms are straightforward (simple offset-based noise sampling). The contribution is primarily empirical rather than methodological.

-- superficial architectural analysis: The paper identifies that models fail but provides no mechanistic explanation/model of why or which architectural components are responsible. Claims about "lack of temporal integration" are descriptive rather than analytical, with only vague suggestions for solutions ("incorporate temporal coherence pathways") and no actionable proposals or preliminary results.

-- incomplete fine-tuning analysis: The fine-tuning experiments claim to demonstrate "fundamental architectural inability" but provide no training diagnostics. Critical questions remain unanswered: What was the training set accuracy? Did the loss decrease during training? Can models recognize training examples after fine-tuning? Without this evidence, it's hard to distinguish between true architectural limitations and experimental design issues, undermining the paper's central claim.

**Questions:**

see my point on "incomplete fine-tuning analysis".

---

### Note · Authors · 2025-11-12

I have read and agree with the venue's withdrawal policy on behalf of myself and my co-authors.